# LiFE-Net: Longitudinal information Fusion for Enhanced lesion detection in unsupervised learning contexts

**Walid Yassine**[1,2]  🆔    WALID.YASSINE@INCEPTO-MEDICAL.COM

[1] *Incepto Medical - France*

[2] *MICS, CentraleSupélec, Université Paris Saclay - France*

**Martin Charachon**[1]    MARTIN.CHARACHON@INCEPTO-MEDICAL.COM

**Céline Hudelot**[2]    CELINE.HUDELOT@CENTRALESUPELEC.FR

**Roberto Ardon**[1]    ROBERTO.ARDON@INCEPTO-MEDICAL.COM

**Editors:** Accepted for publication at MIDL 2025

## Abstract

Accurate detection of liver lesions in longitudinal follow-up is critical for assessing disease progression. Unlike clinical practices that compare multiple time points, most deep-learning approaches treat these time points independently. Existing longitudinal imaging methods, particularly in brain imaging, use strategies like channel-wise concatenation, recurrent architectures, or temporal difference computation. However, these methods might fall short in liver imaging due to challenges like non-rigid motions, anatomical variability, and changes in imaging conditions. To address these challenges, we introduce LiFE-Net, the first framework to integrate longitudinal information from baseline liver CT scans through feature fusion. Our method employs intermediate feature fusion via self-attention mechanisms, leveraging baseline images to incorporate longitudinal information for more accurate predictions. We adopt an unsupervised training approach using synthetic lesions to address the lack of supervised datasets for longitudinal liver tumors. Our results show improvements in detection performance on follow-up images when baseline information is incorporated, with gains in both detection mAP and ROC AUC per exam metrics. An exhaustive ablation study further highlights the impact of baseline image integration, registration quality, and architectural components in achieving these improvements. Our code for LiFE-Net is made publicly available at: https://github.com/walid-yassine/LiFE-Net

**Keywords:** Longitudinal imaging, Liver lesion detection, Deep learning, CT scans.

## 1. Introduction

Liver cancer poses a significant challenge in oncology, ranking as the sixth most frequently diagnosed cancer and the fourth leading cause of cancer-related mortality worldwide (Bray et al., 2018; Tolou-Ghamari and Palizban, 2022). Early detection and longitudinal monitoring of liver tumors are critical for improving patient outcomes and guiding therapeutic decisions. Clinicians typically rely on longitudinal comparisons of imaging data, evaluating changes in tumor size and morphology to assess treatment efficacy and disease progression, as outlined in the RECIST guidelines (Eisenhauer et al., 2009; Forner et al., 2018).

Deep learning methods have emerged as powerful tools for leveraging temporal information in medical imaging. Some existing works treat this task as a tumor-tracking problem (Moltz et al., 2012; Cai et al., 2021; Hering et al., 2021; Tang et al., 2022), identifying lesions in the current scan based on their positions in previous scans. Such methods are

limited to tracking lesions present in both scans without accounting for new or disappearing lesions. In brain imaging, particularly for longitudinal analysis of multiple sclerosis (MS) lesions, public datasets have enabled significant advancements (Danelakis et al., 2018; Zeng et al., 2020; Hammer et al., 2024; Joskowicz et al., 2024; To et al., 2021; Birenbaum and Greenspan, 2017b). These methods often treat each timepoint independently or concatenate scans from different timepoints as input to convolutional neural networks (CNNs) that predict lesion masks in the current image. Some methods use channel-wise concatenation, stacking 2D/2.5D slices (Birenbaum and Greenspan, 2017a; Denner et al., 2021) from multiple timepoints along input channels to process temporal information simultaneously. More advanced methodologies have sought to incorporate temporal dynamics by leveraging pairs of 3D scans. (Szeskin et al., 2023) used a recurrent residual U-Net (R2UNet) model applied to liver longitudinal images. This architecture processes registered scan pairs and uses temporal information by concatenating prior and current 3D scans along the channels. Similarly, (Wu et al., 2023) introduced a framework that segments both new and all lesions in the current scan, integrating heterogeneous dataset annotations from single and dual timepoint scans. While these methods demonstrate promising results, they do not explicitly enforce the model to focus on inter-timepoint differences and risk overemphasizing one scan, leading to performance that resembles single timepoint models (Rokuss et al., 2025). (Rokuss et al., 2025) addressed this issue in brain imaging by proposing a temporal difference weighting method with an explicit architectural bias to emphasize inter-timepoint differences by computing difference feature maps in the encoder. This approach explicitly models temporal dynamics through feature differences and has achieved state-of-the-art (SOTA) performance in longitudinal brain imaging for MS lesion analysis.

However, most of these advancements are concentrated on brain imaging, where datasets and annotations are more readily available. The broader application of such methods to liver imaging remains an open challenge. Longitudinal analysis of liver CT scans faces distinct challenges. Unlike the brain, the liver experiences non-rigid and dynamic deformations, making it challenging to model temporal changes. Additionally, variations in contrast levels and anatomical shifts caused by surrounding organs further complicate the analysis. This makes the direct transfer of methodologies from brain imaging less effective. Moreover, the limited availability of annotated paired scans for longitudinal liver CT studies - primarily due to the absence of publicly available datasets - adds another layer of difficulty.

To address these limitations and challenges, we propose a novel framework better suited for longitudinal liver CT scan analysis. Our approach is designed to answer the following key questions: $(Q_1)$ How can we leverage longitudinal information for improving liver tumor analysis while adhering to clinical practices? $(Q_2)$ What is the impact of incorporating prior information and the quality of input registration on model performance? and $(Q_3)$ Can such approaches overcome the challenge of limited annotations, especially in longitudinal studies where obtaining annotated paired scans over time is more complex than single-scan annotation? In response to these questions, we present the following contributions:

- A novel deep learning framework for longitudinal liver CT scan analysis that leverages temporal information through feature-level fusion to enhance tumor detection.

- Incorporation of explicit architectural mechanisms, such as stochastic masking combined with self-attention, to balance the model's focus across multiple scans and mitigate over-reliance on a single time point.

- A comprehensive evaluation of the proposed framework, showcasing its superior performance in lesion detection over existing SOTA methods.

- Extensive ablation study to understand the impact of baseline image integration, registration quality, and architectural components on the proposed framework.

## 2. Methodology

Let $V : V(x, y, z) \in \mathbb{R}^{H \times W \times D}$ represent a 3D liver CT scan, where $H$, $W$, and $D$ denote the height, width, and depth of the scan, respectively. The liver segmentation mask is denoted by: $S_{Liver} : S_{Liver}(x, y, z) \subset V$. Similarly, $S_{Lesion} : S_{Lesion}(x, y, z) \subset S_{Liver} \subset V$, represents the segmentation mask of the lesions within the liver.

### 2.1. Leveraging Longitudinal Information for Improved Lesion Detection

We propose **LiFE-Net** (Longitudinal information Fusion for Enhanced lesion detection network) to enhance lesion detection in follow-up CT scans by leveraging prior imaging data. Unlike single-timepoint methods, the model leverages both baseline (prior) $V^p$ and follow-up (current) $V^c$ scans to co-learn a temporally informed representation (Fig. 1).

The model processes two types of inputs: the follow-up and baseline inputs. The **follow-up scan inputs** consist of the current volume $V^c$ and its corresponding liver segmentation mask $S_{Liver}^c$. These inputs are concatenated along the channel axis and passed into the encoder of the followup branch $E^c$ (that follows the architecture of an nnU-Net encoder). The **baseline scan inputs** include the prior volume $V^p$, the liver segmentation mask $S_{Liver}^p$, and the lesion segmentation mask $S_{Lesion}^p$ weighted by a softening coefficient $\alpha_{soft}$, which are also concatenated along the channel axis and processed by the encoder of the prior branch $E^p$ (same architecture as $E^c$). Encoded features from $E^c$ and $E^p$ are concatenated along the channel axis and fed into a joint encoder $E^{joint}$ (that also follows the architecture of an nnU-Net encoder). Implementation details of these encoders can be found in Appendix B. The key components of the architecture are described as follows:

1. **Feature Extraction**: Individual encoders $E^p$ and $E^c$ process the baseline and follow-up inputs to generate multi-resolution feature maps $F_{[1:n]}^p$ and $F_{[1:n]}^c$, where $n$ is the number of encoders resolution levels (e.g., if $E^c$ has 4 downsampling blocks $\rightarrow n = 4$; $F_{[i]}^c$ is the feature map from $E^c$ at resolution level $i$ i.e. from $E_{[i]}^c$). The encoders $E^p$ and $E^c$ share the same architecture but have different parameters.

$$F_{[i]}^p = E_{[i]}^p([V^p, S_{Liver}^p, \alpha_{soft} S_{Lesion}^p]), F_{[i]}^c = E_{[i]}^c([V^c, S_{Liver}^c]) \quad \text{for } i \in \{1, \ldots, n\} \quad (1)$$

2. **Feature Fusion with Self-Attention**: Feature maps at the resolution level $n$, $F_{[n]}^p$ and $F_{[n]}^c$, from $E^p$ and $E^c$ are concatenated and passed through a self-attention (SA) mechanism before being fed into the joint encoder $E^{joint}$ to generate the multi-resolution joint feature maps $F_{[1:n']}^{joint}$ of the inputs, where $n'$ refers to the number of

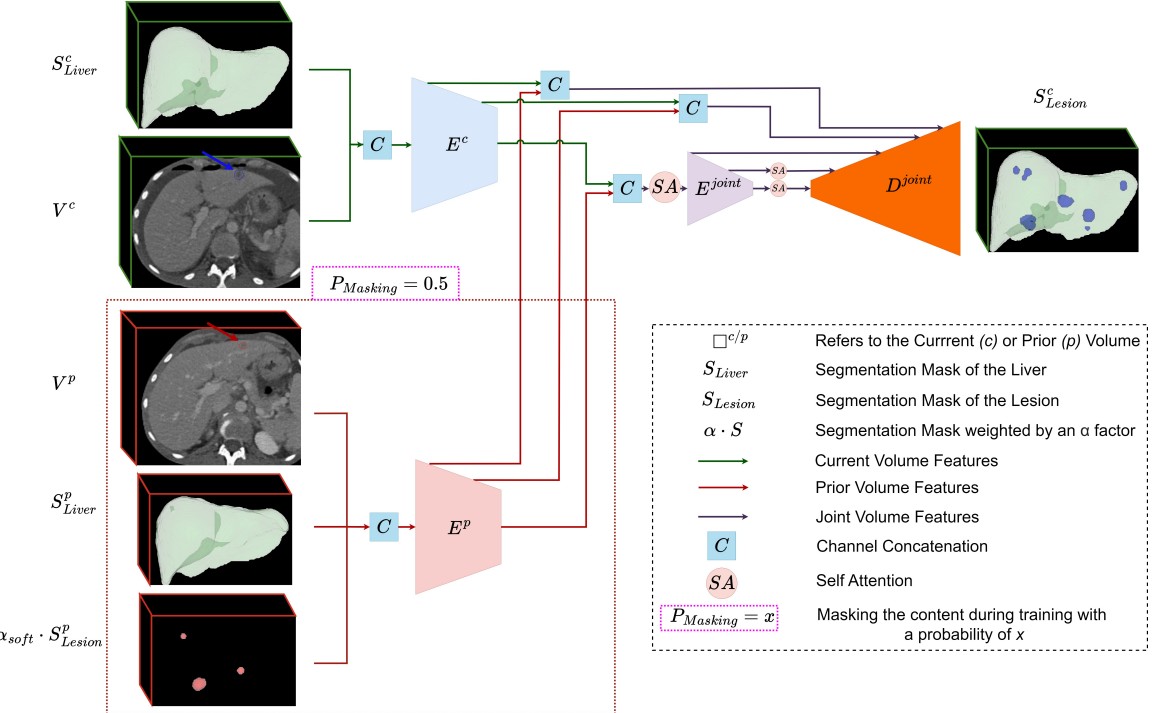

Figure 1: **LiFE-Net architecture overview:** The model processes follow-up inputs ($V^c$, $S^c_{Liver}$) and baseline inputs ($V^p$, $S^p_{Liver}$, $\alpha_{soft} \cdot S^p_{Lesion}$) through separate encoders ($E^c$, $E^p$). Features are fused using a joint encoder with self-attention to capture relevant temporal patterns. A stochastic masking strategy ($P_{masking} = 0.5$) is applied during training. Skip connections integrate multi-resolution features, and the decoder $D^{joint}$ generates the follow-up lesion segmentation mask.

resolution levels of $E^{joint}$. This mechanism allows the model to account for baseline features in follow-up lesion detection, capturing relevant longitudinal patterns.

$$F^{joint}_{[j]} = E^{joint}_{[j]}(SA([F^p_{[n]}, F^c_{[n]}])) \quad \text{for } j \in \{1, \ldots, n'\} \tag{2}$$

3. **Skip Connections and Decoding**: Following nnU-Net principles, skip connections are used to propagate information during decoding. The decoder $D^{joint}$ (nnU-Net decoder architecture) uses the fused representation from $E^p$, $E^c$, and $E^{joint}$ to generate a lesion segmentation mask for the current scan $V^c$. $D^{joint}$ takes as input the feature maps at the resolution level $n'$, $F^{joint}_{[n']}$, from $E^{joint}$ along with the feature maps $F^{joint}_{[1:n'-1]}$ as skip connections. Above the resolution level $n'$, feature maps from $E^p_{[1:n-1]}$ and $E^c_{[1:n-1]}$ are concatenated before being passed as skip connections to the decoder.

$$S^c_{Lesion} = D^{joint}([F^{joint}_{[1:n']}, F^p_{[1:n-1]}, F^c_{[1:n-1]}) \tag{3}$$

4. **Stochastic Masking and Lesion Mask Weighting**: A stochastic masking strategy is employed during training, where all the features maps of the prior branch $F^p_{[1:n]}$ are masked with a probability of $P_{masking} = 0.5$. This encourages the model to perform effectively without prior information, allowing the self-attention mechanism to identify the longitudinal patterns it should leverage from the encodings generated by both $E^p$ and $E^c$ encoders. Additionally, the baseline lesion segmentation mask $S^p_{Lesion}$ is modulated by a weighting factor $\alpha_{soft}$ to reduce over-reliance on prior lesion masks. Without this, the model might default to trivial solutions, replicating $S^p_{Lesion}$ at the output due to the high likelihood of lesions persisting in $S^c_{Lesion}$.

### 2.2. Synthetic Lesion Generation for Longitudinal Data

To overcome the lack of supervised longitudinal datasets, we use a synthetic lesion generation approach based on (Hu et al., 2023), which creates realistic tumors within the liver region $S_{Liver}$ with corresponding ground truth lesion masks $S_{Lesion}$. The process begins by selecting a lesion location within the liver. Initially, the lesion is modeled as an ellipsoid shape, which is then deformed through elastic transformations. Next, a texture is generated from Gaussian noise and adjusted to simulate hypodense lesions with lower Hounsfield Unit (HU) values than liver tissue. Finally, the lesion is embedded into the scan. While our tumor synthesis strategy is based on this approach, we introduced several modifications to improve tumor quality, model performance, and generalization ability on real tumors. Further details on these adaptations can be found in Appendix C. We further adapt this generation framework for longitudinal liver CT scans to ensure both spatial and temporal consistency between baseline and follow-up scans:

- **Spatial Consistency**: Lesions in the follow-up scan are placed near their baseline locations. A translation augmentation of $[0-6mm]$ is applied to the new lesion's center to account for registration errors and liver deformation. This helps correct shifts, especially for subcapsular lesions near the liver boundary, due to the liver's non-rigid structure. Lesions are added before registration to ensure that the transformation affects them as in real cases and prevents bias from the registration model.

- **Temporal Consistency**: Synthetic lesions are designed to mimic realistic clinical behaviors, such as stability, growth, shrinkage, or disappearance over time. Lesion evolution is modeled as follows: 20% of lesions appear in only one scan ($V^c$ or $V^p$), while 60% appear in both. Among lesions present in both scans, 30% remain stable, 30% grow, 10% shrink, and 30% show a mix of stability and growth. To model growing and shrinking lesions, an ellipsoidal shape is initialized near the original lesion (as described in **Spatial Consistency**). The lesion radius is then adjusted: for growth, it is sampled from $[1.3r, 1.7r]$; for shrinkage, from $[0.5r, 0.7r]$, where $r$ is the previous scan's lesion radius. Finally, elastic deformations are applied to lesion shapes to simulate realistic morphological changes.

This synthetic augmentation provides control over tumor characteristics. It also allows the model to learn from various lesion behaviors across time points.

## 3. Experiments

### 3.1. Dataset and Implementation Details

**Training Dataset (Synthetic Tumors):** The training dataset $D_{train}$ consists of 604 **healthy** longitudinal liver exam pairs (baseline and follow-up) from 330 patients, identified as healthy based on clinical reports. Synthetic tumors are added to the scans, resulting in 1,812 follow-up pairs (33% healthy pairs and 67% with synthetic lesions in $V^p$ or/and $V^c$). Further details on tumor generation, including distributions of tumor types, sizes, counts, and other attributes, are in Appendix C.

**Evaluation Dataset (Real Tumors):** The independent evaluation dataset, $D_{eval}$, consists of 192 exams from 83 patients, forming 110 follow-up pairs. Among these, 69% of follow-up scans ($V^c$) contain at least one tumor. For patients with multiple follow-ups, each scan pair is analyzed separately (e.g., a patient with 3 exams $[e_1, e_2, e_3]$ provides 2 pairs: $[e_1, e_2]$ and $[e_2, e_3]$). The mean time between scans is 274 days (range: 8–1141). Baseline scans (40 healthy, 70 pathological) averaged 1.25 lesions per exam (range: 0–40), with a mean lesion diameter of 23.3 mm (range: 6.2–189.0 mm), and a mean lesion volume of 16.05 mL (range: 0.06–448.93 mL). Follow-up scans (34 healthy, 76 pathological) averaged 1.82 lesions per exam (range: 0–46), with a mean diameter of 21.6 mm (range: 6.4–118.1 mm) and a mean lesion volume of 10.1 mL (range: 0.08–255.02 mL). Tumors in the dataset include hepatocellular carcinoma (HCC), metastases, hemangiomas, cysts, and abscesses. All tumors are **real** and manually annotated by two radiology residents. To quantify the model performance across lesions of varying sizes, we consider two evaluation subsets: $D_{eval}^{>10}$ excluding lesions with a diameter $< 10mm$ (as in clinical RECIST guidelines (Eisenhauer et al., 2009)) and $D_{eval}^{>6}$ excluding lesions with a diameter of $< 6mm$.

All images are pre-registered using rigid and non-rigid transformations as described in (Yassine et al., 2025) to align baseline and follow-up scans. During training, the $\alpha_{soft}$ weighting coefficient was fixed at 0.3, selected empirically for optimal performance. Two-fold cross-validation experiments were performed on NVIDIA T4 GPUs, using an Adam optimizer, until convergence, with a composite loss combining focal loss and dice coefficient loss: $\mathcal{L}_{composite} = \mathcal{L}_{focal} + 0.5 \cdot \mathcal{L}_{dsc}$. Further implementation details are in Appendix B. At inference, the $S_{Lesion}^p$ mask is derived from a single-timepoint lesion detection model ($M_{single}$) with a nnU-Net architecture trained on synthetic lesions. To keep the LiFE-Net model agnostic to the detection model, we use an augmented version of the ground truth lesion mask as $S_{Lesion}^p$ during training. This augmentation aims to simulate common errors: false positives are introduced by adding random lesions to the segmentation mask $S_{Lesion}^p$. In contrast, false negatives are simulated by removing lesions from $S_{Lesion}^p$ while keeping them in $V^p$. This strategy makes the model training independent of any specific detection model, improving scalability and generalization. The model's robustness is evaluated with two detection models: $M_1$ (3D nnU-Net) and $M_2$ (ensemble of 3D and 2D nnU-Net models), trained on synthetic tumors. Details on synthetic tumor generation are in Appendix C.

### 3.2. Evaluation Metrics

In detection tasks, key metrics include precision and recall, which measure how many predicted positives are true positives and how many actual positives are correctly identified.

However, these metrics are sensitive to the chosen decision threshold; a **sensitive algorithm** (lower threshold) may be preferred when missing a lesion could have critical consequences, while a **precise algorithm** (higher threshold) minimizes false positives. This summarizes model performance across various thresholds and provides a comprehensive evaluation. Additionally, we assess the ability to classify scans as healthy or pathological, considering a scan correctly classified if at least one lesion is detected. We report the **ROC AUC** at the scan level to differentiate between healthy and pathological scans. We also report the **Sensitivity** of the models at varying false positive rates per exam. All metrics are averaged across annotators, with a mean standard deviation of 0.008 (range:[0.0; 0.03]).

## 4. Results and Discussion

Table 1 compares detection metrics for the evaluation subsets $D_{eval}^{>6}$ and $D_{eval}^{>10}$. Results in parentheses refer to $D_{eval}^{>10}$. The models compared are: (i) Channel Concat: an nnU-Net with concatenated inputs, similar to (Wu et al., 2023), (ii) Diff Weighting: the temporal difference weighting longitudinal nnU-Net (Rokuss et al., 2025), with SOTA performance in brain MS segmentation, (iii) 3D R2U-Net (Alom et al., 2018): a multichannel recurrent residual UNet as used in (Szeskin et al., 2023), (iv) LiFE-Net$^{masked}$: with $P_{Masking} = 1$ during inference (excluding the longitudinal branch); and (v) LiFE-Net: with $P_{Masking} = 0$ during inference (including the longitudinal branch). Additional details on model sensitivity at varying false positive rates can be found in Table 2. Table 3 highlights ablation studies on the impact of registration methods, architectural components, and longitudinal information.

**Leveraging temporal information improves lesion detection.** Our proposed method outperforms the baseline and SOTA approaches, achieving higher detection mAP (0.724 vs. 0.641/0.603/0.691) and ROC AUC (0.865 vs. 0.843/0.803/0.813), highlighting the benefits of including temporal information. In contrast, the Diff Weighting model struggles with liver scans, likely due to the difficulty in achieving accurate registration. This model assumes perfect alignment, which is challenging given the liver variability (e.g., deformations, physiological changes, biliary duct dilation, etc.). These factors complicate the interpretation of feature differences computed in the model's difference weighting block.

**Impact of registration on model performance.** Accurate registration significantly improves model performance across metrics. As shown in Table 3, where different registration levels are considered (no registration, global registration, and local registration), complete registration enhances detection mAP (0.724 vs. 0.683) and ROC AUC (0.865 vs. 0.839) on $D_{eval}^{>6}$. These results highlight registration's critical role in multi-timepoint analysis.

**Contribution of architectural components.** Ablation studies (Table 3) underscore the importance of stochastic masking and the weighting coefficient $\alpha_{soft}$ for model performance. Stochastic masking helps the model integrate information from both time points while balancing features from prior and current data. The self-attention mechanism enables this by focusing on relevant features within the joint encoder. Additionally, soft weighting ($\alpha_{soft} = 0.3$) improves performance by reducing the model's reliance on $S_{Lesion}^p$. Appendix

A presents qualitative examples of the impact of using longitudinal information.

| Model | Detection mAP ↑ | | ROC AUC ↑ | |
|---|---|---|---|---|
| | on $D_{eval}^{>6}$ | on $D_{eval}^{>10}$ | on $D_{eval}^{>6}$ | on $D_{eval}^{>10}$ |
| Channel Concat | 0.641 [0.615;0.645] | 0.673 [0.580;0.701] | 0.843 [0.749;0.849] | 0.867 [0.775;0.908] |
| Diff Weighting | 0.603 [0.542;0.637] | 0.626 [0.552;0.645] | 0.803 [0.702;0.863] | 0.832 [0.721;0.879] |
| 3D R2U-Net | 0.691 [0.623;0.757] | 0.703 [0.635;0.768] | 0.813 [0.718;0.870] | 0.841 [0.771; 0.908] |
| LiFE-Net$^{masked}$ (Ours) | 0.720 [0.664;0.791] | 0.734 [0.644;0.771] | **0.868** [0.774;0.913] | 0.878 [0.798;0.925] |
| LiFE-Net (Ours) | **0.724** [0.672;0.802] | **0.739** [0.689;0.812] | 0.865 [0.773;0.911] | **0.882** [0.809;0.932] |

Table 1: **Quantitative Results:** Comparison of models based on detection mAP and ROC AUC, on $D_{eval}^{>6}$ and $D_{eval}^{>10}$. Values in square brackets represent the 95% confidence interval. The models compared are: (i) Channel Concat (similar to (Wu et al., 2023)), (ii) Diff Weighting (Rokuss et al., 2025), (iii) 3D R2U-Net (Szeskin et al., 2023) (iv) LiFE-Net$^{masked}$ (Ours) with $P_{Masking} = 1$ at inference, and (v) LiFE-Net (Ours) with $P_{Masking} = 0$ at inference.

| Model | Sensitivity↑ | | | |
|---|---|---|---|---|
| | @ Avg FP = 0.2 | | @ Avg FP = 0.5 | |
| | on $D_{eval}^{>6}$ | on $D_{eval}^{>10}$ | on $D_{eval}^{>6}$ | on $D_{eval}^{>10}$ |
| Channel Concat | 0.336 [0.235; 0.475] | 0.409 [0.294; 0.555] | 0.452 [0.329; 0.610] | 0.531 [0.398; 0.685] |
| Diff Weighting | 0.341 [0.204; 0.478] | 0.421 [0.251; 0.556] | 0.418 [0.317; 0.572] | 0.507 [0.398; 0.659] |
| 3D R2U-Net | 0.345 [0.232; 0.459] | 0.424 [0.299; 0.551] | 0.477 [0.312; 0.614] | 0.564 [0.386; 0.726] |
| LiFE-Net$^{masked}$ (Ours) | 0.386 [0.284; 0.553] | 0.471 [0.361; 0.644] | 0.513 [0.412; 0.658] | 0.604 [0.508; 0.739] |
| LiFE-Net (Ours) | **0.429 [0.320; 0.596]** | **0.499 [0.386; 0.672]** | **0.533 [0.430; 0.671]** | **0.618 [0.523; 0.758]** |

Table 2: **Sensitivity Analysis:** Comparison of the models based on sensitivity at two different average false positive rates per exam (0.2 and 0.5), on $D_{eval}^{>6}$ and $D_{eval}^{>10}$. Values in square brackets represent the 95% confidence interval.

**Generalization ability relative to baseline predictions.** The proposed approach enhances baseline predictions for both models, $M_1$ and $M_2$, with greater improvement for $M_1$. For $D_{eval}^{>6}$, incorporating longitudinal data increases mAP by 0.011 for $M_1$ and 0.003 for $M_2$. On $D_{eval}^{>10}$, despite a reduction in mAP for $M_2$ (-0.018), the ROC AUC improves (+0.016), indicating better classification of healthy versus pathological exams. The importance of these metrics may vary depending on the clinical context. Overall, longitudinal information helps in detecting smaller lesions while ensuring reliable exam-level classification.

**Validation on the LiTS dataset.** To assess synthetic tumor generation and generalization ability, we evaluated $M_1$ (3D nnU-Net) and $M_2$ (an ensemble of 3D and 2D nnU-Net models) on the LiTS (Bilic et al., 2022) public dataset (131 scans) with real tumors. $M_2$ achieved a dice coefficient (DSC) of 0.642, outperforming $M_1$ (DSC: 0.581). $M_2$ consistently outperforms $M_1$ on both our in-house and LiTS datasets, highlighting the consistency of this performance gap on both datasets. Compared to prior works, $M_2$ surpasses (Hu et al., 2023), (DSC: 0.598 $\sim$ to $M_1$ performance) which uses a 3D Unet model under similar

synthetic-only conditions but lags behind methods using real annotated data during training, such as (Chen et al., 2024) which uses a Swin UNETR model (DSC: 0.679). These results emphasize the robustness and relevance of the synthetic lesion generation pipeline.

**Limitations and areas for improvement.** While resource-effective, reliance on synthetic lesion generation may not fully capture real clinical variability. Expanding datasets with clinically annotated data could improve generalization ability, as shown in (Chen et al., 2024). Additionally, fixed hyperparameters like $\alpha_{soft} = 0.3$ and $P_{Masking} = 0.5$ might not be optimal across all datasets. Future work could explore adaptive weighting strategies and masking probabilities for enhanced performance.

| Model | Type of Registration | | | Architectural Components | | Metrics | | | |
|---|---|---|---|---|---|---|---|---|---|
| | Liver Cropping | Global Reg | Local Reg | Stochastic Masking | $\alpha_{soft}$ | Detection mAP ↑ | | ROC AUC ↑ | |
| | | | | | | $D_{eval}^{>6}$ | $D_{eval}^{>10}$ | $D_{eval}^{>6}$ | $D_{eval}^{>10}$ |
| **Impact of Registration** | | | | | | | | | |
| LiFE-Net | ✓ | ✗ | ✗ | ✓ | 0.3 | 0.683 | 0.701 | 0.839 | 0.879 |
| | ✓ | ✓ | ✗ | ✓ | 0.3 | 0.713 | 0.730 | **0.871** | **0.888** |
| | ✓ | ✓ | ✓ | ✓ | 0.3 | **0.724** | **0.739** | 0.865 | 0.882 |
| **Impact of Architectural Components** | | | | | | | | | |
| LiFE-Net | ✓ | ✓ | ✓ | ✗ | 1.0 | 0.669 | 0.679 | 0.845 | 0.862 |
| | ✓ | ✓ | ✓ | ✗ | 0.3 | 0.686 | 0.7 | 0.841 | 0.853 |
| | ✓ | ✓ | ✓ | ✓ | 1.0 | 0.699 | 0.710 | 0.863 | 0.878 |
| | ✓ | ✓ | ✓ | ✓ | 0.3 | **0.724** | **0.739** | 0.865 | 0.882 |
| **Adaptability Across Baseline Predictions** | | | | | | | | | |
| $M_1$ | ✓ | - | - | - | - | 0.713 | 0.738 | **0.866** | 0.860 |
| LiFE-Net ($M_1$) | ✓ | ✓ | ✓ | ✓ | 0.3 | **0.724** | **0.739** | 0.865 | **0.882** |
| $M_2$ | ✓ | - | - | - | - | 0.737 | **0.772** | 0.866 | 0.878 |
| LiFE-Net ($M_2$) | ✓ | ✓ | ✓ | ✓ | 0.3 | **0.740** | 0.754 | **0.881** | **0.894** |

Table 3: **Ablation studies:** (1) Impact of registration (Liver Area Cropping, Global Registration, and Local Registration), (2) Impact of architectural variations, including the stochastic masking and the $\alpha_{soft}$ weighting coefficient, and (3)LiFE-Net's ability to improve baseline detection models with varying performance levels. $M_1$ and $M_2$ are standalone detection models, while LiFE-Net is our proposed approach.

## 5. Conclusion

This paper presents LiFE-Net, a framework for analyzing longitudinal liver CT scans that uses feature-level fusion to enhance tumor detection. Our results show that leveraging temporal information improves detection performance, particularly for smaller lesions, compared to baseline and SOTA methods. Key components of LiFE-Net, such as stochastic masking and self-attention, enable effective data integration from multiple time points. This highlights the value of longitudinal information in detecting liver lesions, an area less explored in the literature, and its potential to drive further research in this domain and other applications involving longitudinal imaging. Despite its strengths, the method relies on synthetic lesions, which may not fully capture clinical variability. Future work will focus on expanding clinically annotated datasets and exploring adaptive parameter tuning to improve generalization ability and performance.

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

## Appendix A. Qualitative Results

Figure 2 illustrates the impact of incorporating longitudinal information into liver scan analysis. This is achieved by comparing predictions made using a standalone detection model, $M_1$, with those generated by our proposed approach, LiFE-Net. The figure displays two liver scans: a baseline (prior) scan, $V^p$, and a current (follow-up) scan, $V^c$. The ground truth annotations are highlighted in green. Predictions made by the standalone model, $M_1$, are indicated in red, while those from LiFE-Net are shown in blue.

In case (a), the standalone model detects a false positive, which is successfully eliminated by the longitudinal model (LiFE-Net). In case (b), a lesion missed by the standalone model is correctly identified by LiFE-Net, demonstrating the added value of considering both prior and current scans. In case (c), we show a case of biliary duct dilations inside the liver, where the model appears to be disturbed by these structures, leading to false positives. This is likely due to the fact that biliary ducts also exhibit hypodensity relative to liver tissue, similar to tumors, making them challenging to distinguish.

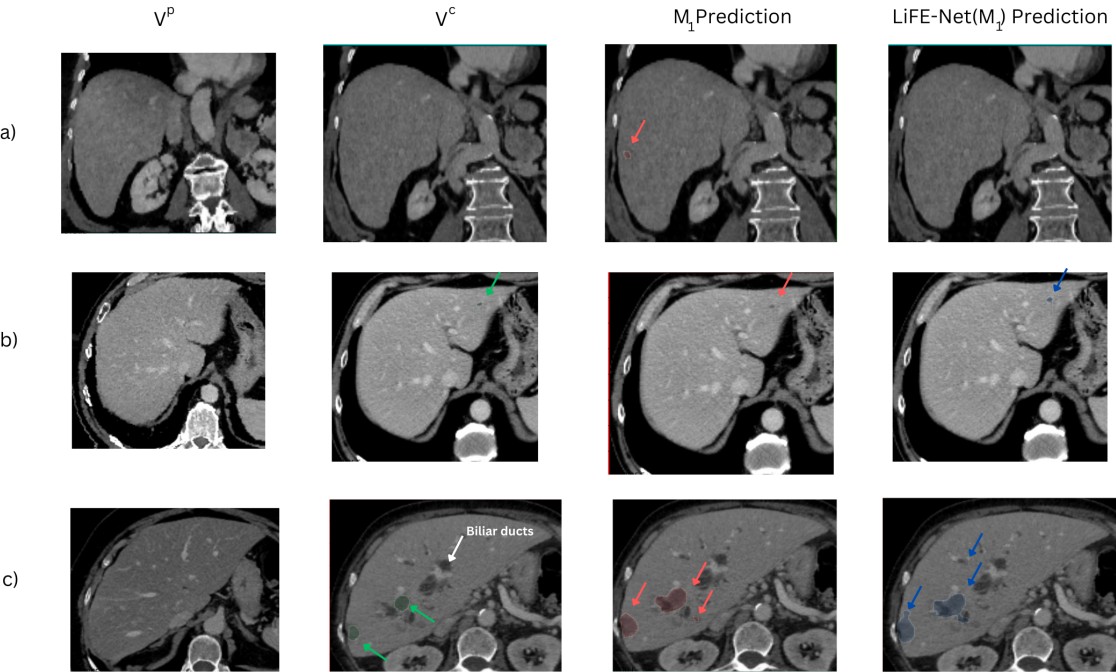

Figure 2: **Impact of using longitudinal information.** Columns (from left to right): prior (baseline) liver scan $V^p$, current (followup) liver scan $V^c$ with ground truth annotation (in green), prediction at $V^c$ with the standalone detection model $M_1$ (in red), and prediction at $V^c$ with our approach LiFE-Net (in blue). (a) A false positive that was eliminated by the longitudinal model. (b) A lesion was missed by the standalone model but was detected through the use of longitudinal information. (c) A case of biliary duct dilations inside the liver, where the model appears to be disturbed by these structures, leading to false positives.

## Appendix B. LiFE-Net: Implementation details

Synthetic tumors are incorporated into the training dataset $D_{train}$, which consists of 604 healthy longitudinal liver examination pairs, to create annotated lesions. For each examination pair $(V^p, V^c)$, two synthetic augmentations are generated by introducing tumors, based on a predefined probability distribution that models longitudinal tumor evolution. This process yields a total of 1,812 follow-up pairs, with 33% representing healthy follow-ups and 67% containing at least one scan ($V^p$ or $V^c$) with lesions. All datasets were collected under GDPR[1] compliance in collaboration with a hospital.

All images are cropped around the liver area and resampled to $(160, 160, 100)$ voxels with an average resolution of $(1.5, 1.37, 2)$ mm along the $(x, y, z)$ axes. Liver segmentation masks are generated using a U-Net model trained on $\sim 1K$ annotated liver masks, achieving a Dice Similarity Coefficient (DSC) of 0.96 on an internal test set and 0.971 on the LiTS dataset (Bilic et al., 2022). All images are pre-registered using rigid and non-rigid transformations as described in (Yassine et al., 2025) to ensure alignment between baseline and follow-up scans. This step minimizes spatial misalignments, aiming to align anatomical structures and lesion positions as closely as possible.

The encoder networks ($E^p$, $E^c$) consist of three down-sampling blocks, starting with 8 filters. The joint encoder ($E^{joint}$) adds two more down-sampling blocks, with an input channel size of 64 (from concatenated encoders) and filters starting at 128. The decoder ($D^{joint}$) includes five up-sampling blocks to generate a lesion segmentation mask ($S^c_{Lesion}$) at the input image resolution. Each down/up-sampling block includes two convolutional layers, with normalization and leaky ReLU activation. During training, the $\alpha_{soft}$ weighting coefficient is fixed at 0.3, a value selected empirically via grid search for optimal performance. Our code for LiFE-Net will be made publicly available at: https://github.com/walid-yassine/LiFE-Net

## Appendix C. Synthetic Tumor Generation

Our methodology for synthesizing tumors is based on existing techniques for liver tumor synthesis, particularly those described in (Hu et al., 2023). However, we have implemented several modifications to enhance both the generation process and the performance of the underlying detection model, as well as its generalization ability to real tumors.

The key improvements to our methodology are as follows:

- **Lesion Size Specification in Millimeters**: The generated tumors are categorized into different classes: tiny, small, large, and mixed (a combination of the previous classes), with a specific radius in millimeters for each class (along with a sampling interval to introduce variability). Characterizing the size of the tumor in millimeters allows for adaptable scaling based on each image's resolution, thereby increasing variability in tumor size during training.

- **Threshold on Lesion Size**: According to RECIST guidelines (Eisenhauer et al., 2009), lesions with an axial diameter smaller than 10 mm cannot be accurately characterized as tumors. Consequently, to comply with radiological standards, we filter

---

1. General Data Protection Regulation

out noisy generations of tiny tumors (especially when tiny lesions undergo elastic deformations that may reduce their maximum axial diameter to below 10 mm). This noise filtering step is quite critical for maintaining the quality of tumor segmentation training, as emphasized by (Yang et al., 2023), where such noisy generations can lead to suboptimal performance of the model. Specifically, we filter out tumors smaller than 7 mm to avoid noisy generations (we use 7 mm instead of 10 mm to leave a safety margin below the threshold typically used in radiological guidelines).

- **Collision Detection with Biliary Ducts**: In addition to avoiding collisions with blood vessels, we propose avoiding collisions with biliary ducts and cavities inside the liver, which appear dark in the acquired image. This ensures that synthetic tumors do not overlap with these critical structures.

- **Adapted Texture Generation**: Instead of using a fixed range of values to sample the texture mean HU value $\mu_t$ for the tumor region, we model it as the mean difference between the liver parenchyma (liver tissue) and the tumor. This allows the tumor texture to adapt to different liver scans, which may have varying HU distributions depending on the phase of acquisition and contrast injection. For instance, exams with late injection times or without contrast typically exhibit lower HU values within the liver. In addition, we use a texture difference sampling strategy conditional on tumor size to minimize noise and ensure plausible tumor appearances. For small tumors, we sample more in the upper part of the texture difference range to reduce sensitivity to image artifacts and minimize false positives, as small tumors that are not markedly hypodense relative to the liver parenchyma can be challenging to detect and may look like artifacts in CT scans. Conversely, we sample more in the lower part of the texture difference range for larger lesions to ensure detection even in the case of subtle texture variations.

- **Contrast Variation in Training Data**: Our training dataset contains cases with varied contrast levels, including those without intravenous contrast and late injection phases. Even though including such cases can be more challenging than considering only well-contrasted ones, they help enhance the model's robustness to image quality and contrast variations.

- **Human in the loop - Parameter Adjustment with Radiologist Input and Post-Training Evaluation**: Involving experts in the parameter tuning process through feedback loops is essential to guarantee relevant synthetic tumors. We perform this feedback process through 2 steps, with a first review phase of the generated tumors that is a common practice. However, the second phase we propose, where radiologists assess the model's detection performance on real tumor data, is essential to uncover any underlying biases and error patterns in the generation process and to achieve good generalization performance on real tumors.

**Main Parametrization:** The tumor generation configuration employs a set of specific parameters to simulate a diverse range of tumors. It includes five distinct tumor types: "tiny," "small," "medium," "large," and "mix," with associated probabilities of occurrence being 0.1, 0.25, 0.3, 0.2, and 0.15, respectively. A tumor of type "mix" samples from each

of the four remaining classes to model different sizes of tumors within the same exam (i.e. a scan with tumors of type "mix" will include tiny, small, medium, and large tumors at the same time.)

For each tumor type, the radius, number, and sigma values ($\sigma_e$ for elastic deformations) are predefined: "tiny" tumors have a mean radius of 6 mm, $\sigma_e$ between 0.7 and 0.9 and their number is uniformly distributed between 5 and 8 tumors, "small" tumors have a mean radius of 8 mm, $\sigma_e$ between 1.0 and 1.25 and their number is uniformly distributed between 4 and 6 tumors, "medium" tumors have a mean radius of 9 mm, $\sigma_e$ between 1.5 and 2.5 and their number is uniformly distributed between 1 and 3 tumors, while "large" tumors have a radius of 14 mm, $\sigma_e$ between 2.5 to 3.5 and their number is uniformly distributed between 1 and 2 tumors. For each class, the tumor radius value is sampled from $[0.75 \times r, 1.25 \times r]$, where $r$ is the mean radius associated with each class. When generating a tumor, a value of $\sigma_e$ is sampled from its associated range, creating various deformations of the initial spherical shape, leading to different shapes and diameters even within the same class. For example, two tumors generated for the "medium" class with different $\sigma_e$ values will have different shapes and radii. Generated tumors are filtered to exclude any tumor whose maximum axial diameter is below 8 mm.

For texture generation, the value $\mu_t$ of the mean difference between the parenchyma and the tumor texture is chosen based on expert feedback, with $\mu_t$ ranging from 30 to 80 HU, meaning that on average, there will be a density difference of 30 to 80 HU.

