# OpenReview forum: "LiFE-Net: Longitudinal information Fusion for Enhanced lesion detection in unsupervised learning contexts"
_MIDL.io/2025/Conference — MIDL 2025 Poster_

### Official Review · Reviewer_if4e · 2025-02-17

**Confidence:** 5
**Preliminary Rating:** 5

**Summary:**

This paper introduces LiFE-Net, a framework that integrates longitudinal information from baseline liver CT scans through feature fusion to enhance lesion detection in follow-up CT scans.
The models were trained on 604 healthy longitudinal liver examination pairs from 330 patients, using synthetic tumors, and tested on an independent dataset comprising 192 exams from 83 patients.
Quantitative results demonstrate that LiFE-Net outperforms SOTA methods.

**Strengths:**

1.The paper presents a novel framework, LiFE-Net, which integrates longitudinal liver CT scans for improved lesion detection. The approach is well-motivated and addresses a key challenge in longitudinal medical imaging.
2. The paper is well-organized, well-written, and easy to follow.
3. Novel longitudinal fusion approach, strong experimental validation, and performance improvements over SOTA.

**Weaknesses:**

1. It is recommended to include qualitative results to visually demonstrate the effectiveness of the proposed method and provide deeper insights into its performance.
2. The study lacks a discussion on real-world validation and generalization.
3. Conducting statistical significance tests and reporting confidence intervals for mAP and AUC is suggested to better illustrate the improvements and provide a clearer understanding of variability across samples.

**Detailed Comments:**

1. Clarification is needed on whether the lesions in the independent evaluation dataset are real, considering that two radiology interns manually annotated the lesion masks.
2. The results section would be enhanced by the inclusion of qualitative visualizations.

**Justification Of The Preliminary Rating:**

This study is well-structured and impactful, but incorporating the suggested improvements would further enhance its strength. Adding qualitative results is recommended, along with a more detailed analysis of small lesions and different lesion sizes, as well as a discussion on real-world generalization.

**Questions To Address In The Rebuttal:**

1. It is recommended to include qualitative results to visually demonstrate the effectiveness of the proposed method and provide deeper insights into its performance.
2. Clarification is needed on whether the lesions in the independent evaluation dataset are real, given that two radiology interns manually annotated the lesion masks.
3. Conducting statistical significance tests and reporting confidence intervals for mAP and AUC is suggested to better illustrate the improvements and provide a clearer understanding of variability across samples.

---

> ### Author Response · Authors · 2025-03-07
>
> Thank you for your constructive feedback. Here is our response to the different points raised in your review:
>
> - *1. It is recommended to include qualitative results to visually demonstrate the effectiveness of the proposed method and provide deeper insights into its performance.*
>
> **Answer:** We appreciate the reviewer’s suggestion. Appendix A includes qualitative examples demonstrating the benefits of our longitudinal model, such as detecting a missed tumor and eliminating a false positive from the standalone model. We have now added an additional challenging case featuring biliary duct dilation, illustrating how the model can be perturbed by structures with similar density to lesions. Thank you for highlighting this point.
>
> - *2. Clarification is needed on whether the lesions in the independent evaluation dataset are real, given that two radiology interns manually annotated the lesion masks.*
>
> **Answer:** We thank the reviewer for this comment.
> We would like to clarify that the lesions in the independent evaluation dataset, $D_{eval}$, are indeed **real tumors**, not synthetic ones. These real lesions were identified and annotated manually by two radiology interns. All the training process is done on synthetic data, but the evaluation and the results shown in the tables and in the qualitative examples are all on **real** tumors.
>
> We recognize that this distinction is crucial and have updated the manuscript to emphasize this point clearly. To improve clarity, we have revised the manuscript by adding two paragraphs to the dataset section in 3.1: one detailing the training dataset, and the other specifically addressing the evaluation dataset. This change ensures the real nature of the lesions in the evaluation dataset is clearly stated. We hope that this addresses the reviewer's concerns and appreciate their input on this aspect.
>
> - *3. Conducting statistical significance tests and reporting confidence intervals for mAP and AUC is suggested to better illustrate the improvements and provide a clearer understanding of variability across samples.*
>
> **Answer:** As suggested by the reviewer, the 95\% confidence intervals are now reported for both the detection mAP and ROC AUC in Table 1. and for sensitivity analysis in Table 2. These help better understand the variability across samples for the different evaluated models. We thank the reviewer for underlining this.

---

> > ### Comment · Reviewer_if4e · 2025-03-14
> >
> > I have no further comments. Thank you for your thorough replies.

---

### Official Review · Reviewer_dQ51 · 2025-02-20

**Confidence:** 5
**Preliminary Rating:** 3
**Final Rating:** 4

**Summary:**

The authors introduce an architecture for fusing annotated prior and current scans along with liver segmentation masks to aid with the detection of liver lesion. They propose individual encoders for both scans and self-attention for subsequent feature fusion as well as stochastic baseline feature masking and lesion mask weighting to steer the training process. The model is trained on a dataset with synthetically generated lesions and evaluated on a manually annotated evaluation dataset, where it performs better than a set of baselines.

**Strengths:**

- The research questions and their clinical relevance are clearly defined. The introduction of longitudinal information is a very relevant topic and the approach described herein is intriguing.

- The paper is mostly easy to follow and flows well, the structure is logical and coherent.

- The authors aim to publicize the accompanying code, which will improve reproducibility.

- The proposed method is compared against the state of the art, strengthening the case for the new architecture.

- The authors simulate false positives and false negatives. This can improve robustness and provide agnosticism from the lesion detection model used for prior lesion masks at inference time, thus improving the method’s applicability.

- The synthetic lesion generation process is improved by introducing a second evaluation phase with trained radiologists at inference time, leading to methodological adjustments of the generation process.

- The authors note that their work is limited by the need to resort to synthesized lesions and by the lack of hyperparameter ablations and aim to improve upon these points in future work.

**Weaknesses:**

- The data that is used for training and testing the model is not described extensively enough, more details especially concerning lesion generation and characteristics of the test dataset would be helpful (see detailed comments). This is crucial as the real-world performance of the model depends on the synthetic dataset being representative.

- The proposed method seems promising as it outperforms the state-of-the-art baselines on the computed metrics. However, it would be interesting whether these improvements hold up to a statistical significance test.

- Sensitivity is critical for clinical applications. The reported metrics provide a summary of the detection performance, but reporting sensitivity for various numbers of false positives could give more detailed insights.

- Some claims about the proposed method are far-reaching. These should be substantiated or rephrased.

    - The claim that soft weighting ensures incorporation of prior image content is not substantiated.

    - The concluding claim that leveraging temporal information improves detection performance particularly for smaller lesions is unsubstantiated. The authors provide neither information about lesion sizes within the used datasets, nor a differentiation about the method’s performance on various lesion sizes.

**Detailed Comments:**

- As mentioned in the section about weaknesses, it would be helpful if the datasets that have been used and the statistical characteristics of the lesions therein would be described in more detail.

    - For the training dataset, how many lesions are added synthetically? What are the lesion sizes associated with the mentioned size classes? How are the percentages for lesion behavior over time determined, as mentioned under the paragraph for temporal consistency?

    - For the test dataset, how many lesions are there in the baseline and follow-up scans? How large are they? How are patients with multiple follow-up scans treated? How much time has passed between baseline and follow-up scans? Which primary tumors are present in the scans?

- The authors mention that two-fold cross-validation experiments were performed. What split was used, and how were the trained models resulting from this cross-validation used? Moreover, it could be interesting to see how statistically robust the performance of the proposed method is, especially compared to the baselines as determined over various dataset splits.

- How exactly does the stochastic masking work? Does it mask all baseline features with the given probability, or just parts of them?

- The impact on the method’s performance of various architecture decisions could be ablated further. This includes the presence of a joint encoder.

- The authors state that their reason for using a translation augmentation during training is to improve robustness towards registration errors and liver deformation. The architecture ablation states that accurate registration is critical for performance. The evaluation dataset however is preregistered both rigidly and non-rigidly. It could be interesting to motivate the aforementioned augmentation by evaluating how large of a registration error with respect to the lesion’s center of gravity remains in the evaluation dataset and how well the model performs with various registration errors.

- Overall, the proposed method seems to perform well on the evaluation dataset. For further work, it would be very helpful to get an understanding about the cases in which the model works well and those where it doesn’t. This requires a more fine-grained evaluation of the model performance, for example with respect to lesion size, lesion behavior, new or disappearing lesions, and singular or clusters of lesions.

- Figure 1 would appear clearer if the image visualization of baseline liver and lesion segmentation masks would be split up into the two inputs they are.

**Justification Of The Final Rating:**

The authors have effectively addressed the identified weaknesses in their rebuttal, enhancing both readability and clarity. The expanded description of the datasets significantly improves the interpretability of the results. Additionally, the inclusion of a sensitivity analysis and confidence intervals allows for a more comprehensive evaluation of the method’s performance. While a few points were not fully addressed in this rebuttal, partly due to time constraints, the authors have indicated their intention to incorporate these improvements in a future journal version, which is acceptable.

**Justification Of The Preliminary Rating:**

The research question is relevant and the paper adds valuable information to the field. However, the work falls short of providing extensive information about the datasets in use and the model’s performance. Introducing this information would make the work introduced herein more applicable and would lead to an upgraded rating.

**Questions To Address In The Rebuttal:**

As mentioned in the section about weaknesses, the question of defining more fine-grained characteristics of the datasets in use and the proposed method’s performance would need to be addressed.

---

> ### Author Response · Authors · 2025-03-07
> **Part 1**
>
> Thank you for your constructive feedback. Here is our response to the different points raised in your review:
>
>
> - *The data that is used for training and testing the model is not described extensively enough, more details especially concerning lesion generation and characteristics of the test dataset would be helpful (see detailed comments). This is crucial as the real-world performance of the model depends on the synthetic dataset being representative.*
>
>
>     - *For the training dataset, how many lesions are added synthetically? What are the lesion sizes associated with the mentioned size classes? How are the percentages for lesion behavior over time determined, as mentioned under the paragraph for temporal consistency?*
>         - *How many lesions are added synthetically?* | **Answer:** The tumor generation configuration includes five distinct tumor types: "tiny," "small," "medium," "large," and "mix," with associated probabilities of occurrence being 0.1, 0.25, 0.3, 0.2, and 0.15, respectively. First, the tumor type is sampled based on this distribution (e.g., 25% probability for small tumors). Then, the number of tumors is determined using a predefined distribution for each type (e.g., small tumors follow a uniform distribution $U[4,6]$, meaning an average of five per exam).
>         - *What are the lesion sizes associated with the mentioned size classes?* | **Answer:** Each lesion type has a predefined range for the number of tumors and tumor sizes, which are sampled stochastically during training. "tiny" tumors have a mean radius of 6 mm, and their number is uniformly distributed between 5 and 8 tumors, "small" tumors have a mean radius of 8 mm, and their number is uniformly distributed between 4 and 6 tumors, "medium" tumors have a mean radius of 9 mm, and their number is uniformly distributed between 1 and 3 tumors, while "large" tumors have a radius of 14 mm, and their number is uniformly distributed between 1 and 2 tumors. For each class, the tumor radius value is sampled from $[0.75 \times r, 1.25 \times r]$, where $r$ is the mean radius associated with each class.
>         - *How are the percentages for lesion behavior over time determined, as mentioned under the paragraph for temporal consistency?* | **Answer:** Temporal behavior (growth, shrinkage, stability) is modeled empirically based on clinical observations from a subset of data and validated with input from a radiologist.
>
>
>         - The exact distributions for tumor type, number, size, and deformations are now clearly detailed in the appendix, with additional clarifications in the revised paper.
>
> - *For the test dataset, how many lesions are there in the baseline and follow-up scans? How large are they? How are patients with multiple follow-up scans treated? How much time has passed between baseline and follow-up scans? Which primary tumors are present in the scans?*
>
> **Answer:** We acknowledge the importance of these questions and have provided a detailed explanation of the evaluation dataset characteristics in the revised paper. The answers to all of these questions are now included in Section 3.1, paragraph "Evaluation Dataset". We thank the reviewer for their valuable feedback.
>
> Here is the updated paragraph:
> "The independent evaluation dataset, $D_{eval}$, consists of 192 exams from 83 patients, forming 110 follow-up pairs. Among these, 69\% of follow-up scans ($V^c$) contain at least one tumor. For patients with multiple follow-ups, each scan pair is analyzed individually (e.g., a patient with 3 exams [$e_1$, $e_2$, $e_3$] will provide 2 follow-up pairs [$e_1$, $e_2$] and [$e_2$, $e_3$]). The mean time between scans is 274 (range: 8–1141 days). Baseline scans include 40 healthy and 70 pathological exams, with a mean of 1.25 lesions per exam (range: 0–40). The mean lesion diameter is 23.3 mm (range: 6.2–189.0 mm), and the mean lesion volume is 16.05 mL (range: 0.06–448.93 mL). Follow-up scans include 34 healthy and 76 pathological exams, with a mean of 1.82 lesions per exam (range: 0–46). The mean lesion diameter is 21.6 mm (range: 6.4–118.1 mm), and the mean lesion volume is 10.1 mL (range: 0.08–255.02 mL). Tumors in the dataset include hepatocellular carcinoma (HCC), metastases, hemangiomas, cysts, and abscesses. All tumors are **real** and manually annotated by two radiology residents."

---

> > ### Author Response · Authors · 2025-03-07
> > **Part 2**
> >
> > - *The concluding claim that leveraging temporal information improves detection performance, particularly for smaller lesions, is unsubstantiated. The authors provide neither information about lesion sizes within the used datasets, nor a differentiation about the method’s performance on various lesion sizes.*
> >
> > **Answer:** The above claim was made based on the reported method’s performance on various lesion sizes ($D_{eval}^{>6}$ and $D_{eval}^{>10}$). To quantify the model performance across lesions of varying sizes, we consider two evaluation subsets: $D_{eval}^{>10}$ excluding lesions with a diameter $< 10mm$ (as in clinical RECIST guidelines and $D_{eval}^{>6}$ excluding lesions with a diameter of $< 6mm$. This claim has thus been made based on the results presented in Tables 1 and 3 and discussed in **Section 4 - Generalization ability relative to baseline predictions**.
> > Previously, in the tables, all the values between parenthesis referred to the performances on the $D_{eval}^{>10}$ subset. Values with no parenthesis report the performance on the $D_{eval}^{>6}$ subset.
> >
> > We think that this aspect is very important and needs to be clearly stated in the paper. As a result, we will move the description of these 2 subsets to the new paragraph **Evaluation dataset** in section 3.1, to make it easier to follow for the reader.
> > All the tables have also been adjusted to add clearly $D_{eval}^{>6}$ and $D_{eval}^{>10}$ as column names to avoid any ambiguity. We thank the reviewer for this input, and we hope it makes the evaluation process and the claim more clear.
> >
> > - *The claim that soft weighting ensures incorporation of prior image content is not substantiated*
> >
> > **Answer:** The original sentence was misplaced and did not accurately convey the intended message. What we meant to express is that during ablation studies where we did not use soft weighting (i.e., $\alpha_{soft} = 1$), the results were worse, and the model tended to simply replicate the lesions from the prior scan, $S^p_{Lesion}$. This was mainly due to the high likelihood that these lesions would persist in $S^c_{Lesion}$, leading the model to adopt a trivial solution. The soft weighting technique was introduced to mitigate this behavior. It allows the model to take both the prior and current inputs into account without copying prior lesions. This clarification has been added to the manuscript and moved to the discussion section. We thank the reviewer for underlining this.
> >
> > - *Sensitivity is critical for clinical applications. The reported metrics provide a summary of the detection performance, but reporting sensitivity for various numbers of false positives could give more detailed insights*
> >
> > **Answer:**  We agree with the reviewer’s suggestion and thank them for highlighting this. To provide more detailed insights, we have added a new table (Table 2 in the revised paper) that reports sensitivity for each model at two values of average false positives per scan (0.2 and 0.5). The results demonstrate that our method surpasses other methods also in terms of sensitivity. Additionally, we have included confidence intervals to better capture the variability across samples for the different methods.
> >
> > - *How exactly does the stochastic masking work? Does it mask all baseline features with the given probability, or just parts of them?.*
> >
> > **Answer:**  Yes, during training, we sample a random variable uniformly between 0 and 1; if the sample value is above $P_{Masking}$ (in our case 0.5, i.e., 50\% of the time), we mask all the features of the prior branch. This clarification has been added to Section 2.1.4 in the revised paper.
> >
> > - *The proposed method seems promising as it outperforms the state-of-the-art baselines on the computed metrics. However, it would be interesting whether these improvements hold up to a statistical significance test.*
> >
> > **Answer:** We appreciate the reviewer’s suggestion to evaluate the statistical significance of the improvements observed. We have now added 95\% confidence intervals for all the computed metrics. This helps better understand the model's performance across samples and accounts for variability, offering a clearer view of the robustness and consistency of the improvements.

---

> > > ### Author Response · Authors · 2025-03-07
> > > **Part 3**
> > >
> > > - *The impact on the method’s performance of various architecture decisions could be ablated further. This includes the presence of a joint encoder*
> > >
> > > **Answer:** We agree that the impact of a joint encoder could be further ablated. In preliminary work, we experimented with separate encoders for baseline and follow-up scans to predict stable, growing, or shrinking lesions. However, this approach yielded suboptimal results, which led us to adopt a joint encoder to better fuse representations. Due to time and space constraints in the rebuttal phase, we are unable to conduct additional ablation studies, but we plan to explore this in a future journal paper.
> > >
> > > - *The authors state that their reason for using a translation augmentation during training is to improve robustness towards registration errors and liver deformation. The architecture ablation states that accurate registration is critical for performance. The evaluation dataset however is preregistered both rigidly and non-rigidly. It could be interesting to motivate the aforementioned augmentation by evaluating how large of a registration error with respect to the lesion’s center of gravity remains in the evaluation dataset and how well the model performs with various registration errors.*
> > >
> > > **Answer:** We agree that further analysis of the model's performance under varying registration errors would be valuable, including training without the applied transformations to assess its impact on the evaluation dataset. This will be considered in future ablation studies.
> > > Regarding the sentence: <u>*"The evaluation dataset however is preregistered both rigidly and non-rigidly."*</u>: In the ablation studies (Table 3 - Impact of Registration), we evaluate the performance of the model on different registration levels (no registration, global registration, and local registration). For example, in the second row of the table, only cropping around the liver area and global registration are applied to both the training images and evaluation dataset during the inference. This ensures that the impact of registration is properly reflected at inference time.
> > > We have added further clarification on this in Section 4 - Impact of registration on model performance.
> > >
> > > - *Overall, the proposed method seems to perform well on the evaluation dataset. For further work, it would be very helpful to get an understanding about the cases in which the model works well and those where it doesn’t. This requires a more fine-grained evaluation of the model performance, for example with respect to lesion size, lesion behavior, new or disappearing lesions, and singular or clusters of lesions.*
> > >
> > > **Answer:** The model's performance concerning lesion size is now reported in the paper through evaluations on $D_{eval}^{>6}$ and $D_{eval}^{>10}$ (cf. response above). We agree that evaluating performance based on lesion behavior (appearing, stable, growing, shrinking) would provide valuable insights, particularly for a clinical evaluation study focused on these aspects. This analysis could be included in a future journal paper, following the annotation of a more extensive evaluation dataset that includes lesion diameter measurements, evolution types, and other relevant characteristics beyond segmentation.
> > >
> > > - *Figure 1 would appear clearer if the image visualization of baseline liver and lesion segmentation masks would be split up into the two inputs they are.*
> > >
> > > **Answer:**  We fully agree with the reviewer. Due to space constraints, we initially combined the inputs, but we acknowledge that the figure should clearly convey the input structure. With the updated page limit, we have modified the figure to explicitly show the two separate inputs. We appreciate the reviewer’s feedback.

---

> > > > ### Comment · Reviewer_dQ51 · 2025-03-11
> > > >
> > > > I appreciate the authors’ detailed and thorough response in clarifying the open points. Their efforts have significantly improved the manuscript, and I have no further discussion points. Thank you for your thoughtful replies.

---

### Official Review · Reviewer_esvV · 2025-02-21

**Confidence:** 4
**Preliminary Rating:** 3
**Recommendation:** Poster
**Final Rating:** 4

**Summary:**

This paper proposes a method named LiFE-Net that uses temporal feature fusion for tumor detection from longitudinal liver CT scans. The method outperforms several baselines on an in-house and a publicly available Liver Tumor Segmentation (LiTS) dataset.

**Strengths:**

1. One of the main strengths of the paper is that longitudinal CT scans are used for lesion detection. In most of the clinical scenarios, there are follow-up scans. Existing methods fail to incorporate these scans for downstream tasks.
2. The experimental design is simple yet innovative. Given, that there is a constraint of longitudinal CT scans for training, this work proposes a unique way of generating synthetic lesion. However, the clinical applicability of the method is doubtable. (Weakness Point 1)
3. The work demonstrates superior performance for different baselines and carries out detail ablation analysis showcasing the robust experimental set-up.
4. The paper is easy to read and follow.

**Weaknesses:**

1. The main weakness of the proposed method is the synthetic dataset used for training. The authors did specify how the synthetic tumors are added (Appendix C). But several questions arise from the clinical applicability standpoint.
    1.a. Did they take the anatomical constraints into consideration? For example, did they take into consideration that the tumor size may change across different timepoints. If yes, then how?
    1.b. How many tiny, small, large, mix tumor types are there? What is mix category? How are they injected into the longitudinal scans? These factors are critical in tumor segmentation.
2. Despite a detailed experimental set-up, the proposed architecture lacks novelty. There are several methods in non-medical imaging domains that have proposed innovative architectural design to incorporate longitudinal data.
3. The authors do not compare with state-of-the-art methods for LiTS such as nnU-Net, SwinUNETR etc.

**Detailed Comments:**

1.	In 2.1.1, what are the feature extractors?
2.	Are $E^p$ and $E^c$ the same? What is $E_{joint}$? Any reason for representing $p$ and $c$ as superscript and $joint$ as subscript?
3.	What is $D_{joint}$?
4. Section 2.1 can be explained in more simple terms by using mathematical equations. It is a bit difficult to follow the subsections 2.1.1, 2.1.2 and 2.1.3.

**Justification Of The Final Rating:**

The authors have thoroughly addressed my key concerns and updated the manuscript accordingly. My primary inquiry about the novelty of the method has been effectively resolved in Point 2 of Part 1 of their response.

**Justification Of The Preliminary Rating:**

The proposed method uses a simple but unique approach to train a lesion detection architecture with synthetic tumors injected in CT scans across different timepoints from healthy patients. However, I have concerns regarding the novelty in the method and its clinical applicability. Please see the detailed comments in weakness and comments section of the review.

**Questions To Address In The Rebuttal:**

The score driving critiques are Weakness point 1 and 2. Please address them in-detail.

---

> ### Author Response · Authors · 2025-03-07
> **Part 1:**
>
> Thank you for your constructive feedback. Here is our response to the different points raised in your review:
>
> - *1.a. Did they take the anatomical constraints into consideration? For example, did they take into consideration that the tumor size may change across different timepoints. If yes, then how?*
>
> **Answer:** Yes, The anatomical constraints of changing lesions are indeed taken into consideration during the tumor generation process. Synthetic lesions are designed to mimic realistic clinical behaviors, including stability, growth, shrinkage, and disappearance. Lesion evolution is modeled such that 20% of lesions appear in only one scan, while 60% appear in both. Among those present in both scans, 30% remain stable, 30% grow, 10% shrink, and 30% exhibit a combination of stability and growth.
>
> To generate growing and shrinking lesions, we initialize an ellipsoid near the center of the original lesion (*Section 2.2 - Spatial Consistency*) and adjust its radius accordingly. The new radius is sampled from the range [1.3r, 1.7r] for growing lesions and [0.5r, 0.7r] for shrinking lesions, where $r$ is the radius in the prior scan $V^p$. The lesion shapes are then subjected to controlled deformations to reflect realistic clinical scenarios. A more detailed clarification of this aspect has been added to "Section 2.2 - Temporal Consistency", in the revised version of the paper.
>
> - *1.b. How many tiny, small, large, mix tumor types are there? What is mix category? How are they injected into the longitudinal scans? These factors are critical in tumor segmentation.*
>     - *How many tiny, small, large, mix tumor types are there"* | **Answer:** The tumor generation configuration includes five distinct tumor types: "tiny," "small," "medium," "large," and "mix," with associated probabilities of occurrence being 0.1, 0.25, 0.3, 0.2, and 0.15, respectively. First, the tumor type is sampled based on this distribution (e.g., 25% probability for small tumors). Then, the number of tumors is determined using a predefined distribution for each type (e.g., small tumors follow a uniform distribution $U[4,6]$, meaning an average of five per exam).
>     - *What is mix category?* | **Answer:**  The "mix" category samples from all four types, simulating scans where multiple tumor sizes coexist. If "mix" is selected, the tumor generation process applies to all four types within the same scan.
>     - *How are they injected into the longitudinal scans?* | **Answer:** For injection into scans, once the lesion shape and position are determined, the texture is generated from Gaussian noise and adjusted to simulate hypodense lesions with lower Hounsfield Unit (HU) values than liver tissue. The lesion is then embedded by overlaying it onto the scan and applying smoothing filters for realistic integration. We didn't add a lot of details for the process of overlaying the lesion into the scan as it follows the same process described in *(Hu et al. 2023)*, whose implementation is available on their [GitHub](https://github.com/MrGiovanni/SyntheticTumors).
>
>
> Additional details on tumor types, count distribution, and the "mix" category have been added to Appendix C. We thank the reviewer for highlighting this.
>
> Finally, note that LiFE-Net does not rely on the tumor generation framework; it can be trained in a supervised manner if annotated real tumors are available. The synthetic tumor generation approach is primarily used to overcome the challenge of limited annotated datasets.
>
> - *2. Despite a detailed experimental set-up, the proposed architecture lacks novelty. There are several methods in non-medical imaging domains that have proposed innovative architectural design to incorporate longitudinal data.*
>
> **Answer:** We acknowledge the reviewer’s concern regarding the novelty of the proposed architecture. It is true that intermediate feature fusion has been previously explored in various domains, such as in remote sensing, where longitudinal studies are commonly used for change detection in satellite imagery. For instance, the survey *(Parelius et al. 2023; doi=10.3390/rs15082092)* presents multiple deep learning frameworks for such applications, including approaches such as image differencing and difference weighting—similar to some of the methods we compare against in our paper. Additionally, some works have explored fusing features before passing them to a decoder network.
>
> Nevertheless, our proposed approach has two key novelties, which to the best of our knowledge, have not been explored in previous works.
> a.  **Architectural:** Self-attention mechanism applied in the context of longitudinal studies on the joint features $F^{joint}$.
> b.  **Training:** The stochastic masking of the prior input branch.
> These two elements improve model performance, as reported in the ablation studies. These aspects differentiate our approach from existing methods and contribute to the novelty of our work.

---

> > ### Author Response · Authors · 2025-03-07
> > **Part 2**
> >
> > - *3. The authors do not compare with state-of-the-art methods for LiTS such as nnU-Net, SwinUNETR, etc.*
> >
> >     - **Regarding the nnU-Net model:** We report the results of both the $M_1$ (a 3D nnU-Net) and $M_2$ (an ensemble of 3D and 2D nnU-Net models) models on the LiTS public dataset (131 scans). We also compare with the performance reported in *(Hu et. al, 2023)*, which also uses a 3d UNet model trained on synthetic-only tumors.
> >     - **Regarding the SwinUNETR model:** We also compare the performance of the $M_1$ and $M_2$ models on the LiTS dataset against the method presented in *(Chen et al. 2024)*, which uses a SwinUNETR architecture. The specification of this architecture was previously missing from the manuscript. Although this model achieves higher results than $M_1$ and $M_2$, it was trained on real annotated tumors.
> >     - **For other approaches:** We report the highest-performing model from *(Chen et al. 2024)* for comparison and to conserve space in the paper. Further details on the performance of other models can be found in *(Chen et al. 2024)*.
> >
> >
> > In the revised version of the paper, we have clarified the architecture details of $M_1$, $M_2$, and SwinUNETR in "Section 4, Subsection: Validation on the LiTS dataset" to remove any ambiguity. We thank the reviewer for pointing this out.
> >
> > - *1. In 2.1.1, what are the feature extractors?*
> >
> > **Answer:** The feature extractors are the encoders $E^p$ and $E^c$, which are 3D encoders with the same architecture as nnU-Net encoders (the architecture details are presented in the "Implementation Details" section of the Appendix B.). To improve clarity, we have added more details to section 2.1.1 regarding this.
> >
> > - *2. Are $E^p$ and $E^c$ the same? What is $E_{joint}$? Any reason for representing $p$ and $c$ as superscripts and $joint$ as a subscript?*
> >
> >     - *Are $E^p$ and $E^c$ the same?* | **Answer:** The encoders share the same architecture but do not share the same parameters. Preliminary experiments showed that sharing parameters between encoders led to suboptimal performances, which is why separate parameters are used for $E^p$ and $E^c$ in subsequent experiments.
> >     - *What is $E_{joint}$?* | **Answer:** $E^{joint}$ is a third encoder with the same architecture as nnU-Net encoders. It starts at the resolution where $E^p$ and $E^c$ end (architecture details in implementation details in Appendix B).
> >     - *Any reason for representing $p$ and $c$ as superscripts and $joint$ as a subscript?* | **Answer:**  There is no specific reason for using superscripts for $p$ and $c$ and a subscript for $joint$. The representations will be standardized.
> >
> >
> > To enhance clarity and eliminate ambiguity, the architecture of $E^p$, $E^c$, and $E^{joint}$ will be mentioned at the beginning of the methodology section, along with a more precise mathematical representation of the generated features. Additionally, the naming of the modules has been standardized in the revised version. We thank the reviewer for pointing this out.
> >
> > - *3. What is $D_{joint}$?*
> >
> > **Answer:** $D^{joint}$ follows the same architecture as an nnU-Net decoder. It takes input from the features extracted by $E^{joint}$, as well as those from $E^p$ and $E^c$ through skip connections, to generate the predicted tumor mask. A clearer specification of the $D^{joint}$ architecture, a more precise mathematical representation, and a standardization of its naming to $D^{joint}$ have been added in the revised version.
> >
> > - *4. Section 2.1 can be explained in more simple terms by using mathematical equations. It is a bit difficult to follow the subsections 2.1.1, 2.1.2 and 2.1.3.*
> >
> > **Answer:**  To improve clarity and readability, we have added mathematical equations to subsections 2.1.1, 2.1.2, and 2.1.3. These equations provide a concise representation of the methodology, making it easier to follow the described processes. We thank the reviewer for this valuable suggestion.

---

> > > ### Comment · Reviewer_esvV · 2025-03-15
> > >
> > > Thank you for your detailed rebuttal. I have no further questions or comments.

---

### Author Rebuttal · Authors · 2025-03-07

**Rebuttal:**

We thank the reviewers for their constructive feedback. We have carefully considered all their comments and provided detailed responses addressing each question. **The revised paper has been uploaded as part of the rebuttal-supplementary materials, as requested by the conference.** All modifications are highlighted for clarity in the revised version of the paper.

**Summary of Key Additions:**

- Expanded details on synthetic tumor generation and training dataset characteristics. *[as underlined by Reviewer esvV (1.a and 1.b) and Reviewer dQ51]*
- Comprehensive description of the evaluation dataset, including tumor distribution (numbers, types, sizes) and emphasizing the fact that these are real tumors and not synthetic, with two evaluations based on different tumor sizes. *[as underlined by Reviewer dQ51 and Reviewer if4e]*
- Sensitivity analysis at varying average false positive rates across methods. *[as underlined by Reviewer dQ51]*
- Computation of confidence intervals for reported metrics. *[as underlined by Reviewer dQ51 and Reviewer if4e]*
- Formatting methodology's subsections 2.1.1, 2.1.2 and 2.1.3 via mathematical formatting for more clarity . *[as underlined by Reviewer esvV (4)]*
- Inclusion of a third qualitative result case illustrating a challenging case for the model. *[as underlined by Reviewer if4e]*

**Supporting Material:**

/attachment/443a69d0c64466a12d625dcd1a318d7102b84d12.pdf

---

### Meta-Review · Area_Chair_dS1D · 2025-03-22

**Recommendation:** Accept (Poster)
**Confidence:** 5

**Metareview:**

Metareviewer decision:
Accept

Comments:
All reviewers found the proposed method to be novel and the results promising.

Weaknesses:
The lack of statistical analysis is concerning. The mere addition of 95% CI is not enough. From Tables 1, 2 and 3, the detection performance of a state of the art model like nnU-Net is less than 2% compared to Life-Net. A statistical test on the ROC-AUC (adjusted for multiple comparisons) would clearly identify if the performance is significant compared the different models tested.